# Ethanol- and PARP-Mediated Regulation of Ribosome-Associated Long Non-Coding RNA (lncRNA) in Pyramidal Neurons

**DOI:** 10.3390/ncrna9060072

**Published:** 2023-11-17

**Authors:** Hooriyah S. Rizavi, Hannah E. Gavin, Harish R. Krishnan, David P. Gavin, Rajiv P. Sharma

**Affiliations:** 1Department of Psychiatry, University of Illinois at Chicago, Chicago, IL 60612, USA; hrizavi@uic.edu (H.S.R.); hannahelizabethgavin@gmail.com (H.E.G.); 2Jesse Brown Veterans Affairs Medical Center, Chicago, IL 60612, USA; 3Center for Alcohol Research in Epigenetics, Department of Psychiatry, University of Illinois at Chicago, Chicago, IL 60612, USA; harishrk@uic.edu

**Keywords:** ribosome-bound lncRNA, alcohol use disorder, TRAP, PARP inhibitor, prefrontal cortex, ABT-888, TRAP-sequencing

## Abstract

Although, by definition, long noncoding RNAs (lncRNAs) are not translated, they are sometimes associated with ribosomes. In fact, some estimates suggest the existence of more than 50 K lncRNA molecules that could encode for small peptides. We examined the effects of an ethanol and Poly-ADP Ribose Polymerase (PARP) inhibitor (ABT-888) on ribosome-bound lncRNAs. Mice were administered via intraperitoneal injection (i.p.) either normal saline (CTL) or ethanol (EtOH) twice a day for four consecutive days. On the fourth day, a sub-group of mice administered with ethanol also received ABT-888 (EtOH+ABT). Ribosome-bound lncRNAs in CaMKIIα-expressing pyramidal neurons were measured using the Translating Ribosome Affinity Purification (TRAP) technique. Our findings show that EtOH altered the attachment of 107 lncRNA transcripts, while EtOH+ABT altered 60 lncRNAs. Among these 60 lncRNAs, 49 were altered by both conditions, while EtOH+ABT uniquely altered the attachment of 11 lncRNA transcripts that EtOH alone did not affect. To validate these results, we selected eight lncRNAs (Mir124-2hg, 5430416N02Rik, Snhg17, Snhg12, Snhg1, Mir9-3hg, Gas5, and 1110038B12Rik) for qRT-PCR analysis. The current study demonstrates that ethanol-induced changes in lncRNA attachment to ribosomes can be mitigated by the addition of the PARP inhibitor ABT-888.

## 1. Introduction

Investigations into gene regulatory abnormalities related to alcohol consumption have been intensely studied over the past several decades. Alcohol use disorder (AUD) affects about 10% of adults in the United States (SAMHSA, 2018). Prior studies have documented the important role of the prefrontal cortex (PFC) excitatory pyramidal neurons in the reinforcing effects of alcohol [1,2,3]. While coding genes have received much attention over the years, most RNA transcripts are not translated into proteins [4]. We earlier reported that inhibiting the enzymatic activity of Poly-ADP Ribose Polymerase (PARP), a protein known to play a role in DNA demethylation, histone modifications, and post-translational modifications of transcription factors, can reverse gene expression changes caused by alcohol administration and reduces alcohol-drinking behavior in mice [5,6]. These initial experiments were designed to avoid behavioral or contingent variability and sought to establish a primary pharmacological response of ethanol on ribosome attachment. The objective is to establish contingent parameters that may influence ribosomal attachment. We have recently reported on the effects of ethanol (EtOH) and PARP on the attachment of ribosomes to mRNA transcripts of protein-coding genes [7]. In the current study, we extend our investigations to the effects of EtOH and PARP inhibition on long non-coding RNAs (lncRNA) in the brain. Our prior study demonstrated the effect of EtOH exposure on ribosome-bound mRNA transcripts and showed that ABT-888 can reverse the effects of EtOH. Moreover, we identified the enrichment of the Insulin Receptor Signaling pathway in the pool of ethanol-regulated and PARP-reverted ribosome-bound transcripts. The entire dataset utilized in both studies, including the processed files, is accessible in the GEO repository under accession number GSE227947.

Accumulating evidence indicates that lncRNAs play essential roles in neuropsychiatric disorders, including addiction and psychosis [8,9,10,11]. LncRNAs are transcripts longer than 200 nucleotides that are, by definition, not translated into proteins [12]. LncRNAs display a higher cell-type specificity than protein-coding genes, and approximately 40% of currently identified lncRNAs are expressed in the brain [13,14]. In the nucleus, lncRNAs can enhance or repress transcription of local or distal genes through various mechanisms, such as recruiting or inhibiting the binding of regulatory proteins to gene promoters, directly binding to RNA polymerase II (Pol II), and acting as scaffolds for chromatin-modifying complexes [15,16,17]. The function of lncRNAs in the cytosol is less established, with several studies observing their association with ribosomes. An earlier *in silico* analysis predicts that 48.16% of lncRNAs in mice can interact with ribosomes (up to 61% in the hippocampus) [18]. When associated with ribosomes, lncRNAs can repress or enhance the translation of mRNA transcripts through the formation of RNA–RNA hybrids, targeting them for nonsense-mediated decay (NMD), and have segments of their transcripts be translated into small peptides (sPEPs) [19,20,21,22,23]. These sPEPs can in turn interfere with miRNA and antisense lncRNA and independently regulate gene transcription [24], and directly repress the translation of mRNA [23,25,26].

Ribosome purification combined with unbiased RNA sequencing (TRAP-Seq; Translating Ribosome Affinity Purification followed by RNA sequencing) identifies transcripts bound to ribosomes [19,27,28,29,30]. In the current study, we bred two transgenic mouse lines to obtain GFP-tagged ribosome expression controlled by the CAMKIIα promoter. We investigated the effects of ethanol (EtOH) and the co-administration of EtOH and PARP inhibitor (ABT-888) compared to saline (control condition, CTL) on lncRNA attachment to ribosomes. To supplement the RNA-Seq analysis, we parsed lncRNA identities from the *in silico* analysis of Zeng et al., 2018, who provide lncRNA identities with sequences consistent with ribosomal affinity and that contain peptide-coding potential [18].

## 2. Results

### 2.1. Effects of EtOH and EtOH+ABT-888 over CTL on Ribosome-Attached lncRNAs Measured by RNA-Seq

To answer the question as to whether EtOH with or without supplemental ABT-888 influenced the attachment of lncRNAs to ribosomes, we examined the number of lncRNAs with significant changes compared to the CTL condition by surveying our TRAP-Seq results. A schematic representation of differentially expressed lncRNAs is shown in Figure 1. EtOH altered the ribosome attachment of 107 lncRNAs, while the addition of ABT-888 to EtOH (EtOH+ABT) produced changes in the attachment of 60 ribosome-attached lncRNA transcripts compared to CTL. Of these 60 transcripts, 49 were altered by both the EtOH and EtOH+ABT conditions, and 11 were unique to the EtOH+ABT condition. The difference in the ribosome-bound lncRNA expression profiles between the CNTL (CNT), EtOH, and EtOH+ABT (EA) is illustrated in the Principal Component Analysis (PCA) plot of the normalized log CPM (based on the DESeq2 method) shown in Figure 2A. The individual samples of CNTL and EtOH are clustered well away from each other, indicating a clear difference of differential gene expression between the two groups. Although the EtOH+ABT replicates do not show a clear clustering, on PC1, three of the four replicates show a trend of shifting “closer” to the CNTL group, indicating a change in expression profile of the EtOH treated after the administration of ABT-888. Furthermore, volcano plots were generated by computing the negative logP (−logP) against the log2 fold change (FC) values and evaluating the distribution of lncRNAs perturbed by EtOH with or without supplemental ABT-888 compared to the CTL condition. The data are presented in two volcano plots (Figure 2B,C) highlighting those lncRNAs with the largest and most significant changes in the TRAP-Seq data. Figure 2B presents the effects of EtOH compared to CTL (EtOH/CTL), and Figure 2C presents the effects of EtOH+ABT compared to CTL (EtOH+ABT/CTL). In general, EtOH produced far more changes in lncRNA attachment than EtOH+ABT. It is clear that the effects of EtOH by itself are ‘constricted’ or reversed after the administration of ABT-888. For example, by itself, EtOH appears to affect a broad distribution of lncRNA molecules bound to ribosomes (compared to CTL) with −logP values in the range of 10, and a wide FC spread (−4 to +3.5) (Figure 2B), which would imply that EtOH by itself results in an active adjustment of the lncRNA ribosomal attachment. Alternately, examining Figure 2C indicates there are no genes that achieve a −logP value greater than 4, which is a clear shift from the effects of EtOH (Figure 2B). ABT-888 constricts the range of ribosomal-bound lncRNA expression, with -logP now less than 4 and a constricted range of FC (−2 to +3). Taken together, the effects of ribosomal attachment are approximately greater in EtOH than in the EtOH+ABT condition, as summarized by the ‘constricting’ −logP and FC values across the two distributions. To conduct the analyses in Figure 2, we included all 308 lncRNAs that passed our QC filter across the samples collected. Given the small sample size and low count levels typical of lncRNA results in RNA-Seq experiments, we identified and validated the expression of eight lncRNA molecules by qRT-PCR (Figure 3) selected from the Zeng et al., 2018, analysis (elaborated below), and tagged them in this RNA-Seq-derived volcano plot to monitor the −logP/FC distribution (Figure 2B,C).

### 2.2. qRT-PCR Validation of Putative Protein-Coding lncRNA

We next listed all lncRNAs within our 308 lncRNA transcripts that received an RAI score in the supplemental data provided by Zeng et al., 2018 (described here in methods) [18], and performed a *t*-test for average normalized transcript counts between the EtOH and CTL conditions. We then selected eight molecules (Table 1) that had significant differences in ribosomal association between the EtOH and CTL conditions (Figure 4) and validated their expression levels using qRT-PCR (Figure 3). For validation, we included samples from mice treated with ONLY ABT-888 but that had not been sent for TRAP-Seq to measure any effect of ABT-888 by itself and were treated in concordance with the EtOH+ABT-888 time point (the experimental design scheme is detailed in Section 4.2 and shown in Figure 5). Utilizing the same RNA samples sent for sequencing, we see that the increased attachment of the eight lncRNAs with EtOH administration is also reflected in the significant upregulation of their transcript level as measured using qRT-PCR (Figure 3). Similarly, with the addition of ABT-888, we observed a “normalizing” effect in Gas5, Sng17, miR124, and 1110038B12Rik, which agrees with the TRAP-Seq results (Figure 4). Furthermore, we found that administering ABT-888 alone had no significant effect on any of the lncRNAs measured, indicating that PARP inhibition may be dampening the effect of EtOH.

**Table 1 ncrna-09-00072-t001:** LncRNAs selected for validation.

LncRNA	Protein-Coding Potential	Known Function
MIR124-2hg	A0	Chr 3: MIR124-2 host gene is the most abundant brain-specific miR, promotes microglial quiescence, inhibits microglial and astrocyte activation, and suppresses experimental autoimmune encephalomyelitis by deactivating macrophages via the CEBPA/C/EBP-a-SPI1/PU.1 pathway.
5430416N02Rik	A0	Chr 5: lncRNA 5430416N02Rik locus regulates Mid1 expression through 3D chromatin organization. Klf4 and CTCF bind to 5430416N02Rik RNA [31]. Knockout of the 5430416N02Rik locus reduces the proliferation rate of embryonic stem cells (ESCs).
Snhg17	A2	Chr 2: Small nucleolar RNA host gene 17 is expressed in nervous system, acts as a sponge for multiple miRNAs, such as miR-23a-3p, miR-485-5p, and miR-361-3p, promoting tumor progression in various malignancies.
Snhg12	A3	Chr 4: Small nucleolar RNA host gene 12 is an oncogene that promotes tumor formation through cell cycle, invasion, metastasis, and apoptosis. It also affects the Wnt-β-catenin and MLK3/IκB/NF-κB pathway, Notch-1 signaling pathway, and PI3K/AKT pathway, and serves as a competing endogenous RNA (ceRNA), similar to a miRNA sponge, wherein it attenuates the effects of miRNAs, often leading to cancer progression.
Snhg1	A3	Chr 19: Small nucleolar RNA host gene 1 has a role in synapse formation. It promotes cell proliferation via upregulating β-catenin, c-Myc, and cyclin D1 pathways. It negatively regulates tumor suppressor genes such as tumor protein p53. SNHG1 could directly interact with Polycomb Repressive Complex 2 (PRC2) and modulate the histone methylation of promoter of Kruppel-like factor 2 (KLF2) and Cyclin-dependent kinase inhibitor 2B (CDKN2B) in the nucleus.
MIR9-3hg	A3	Chr 7: LncRNA with a role in immune function. It can maintain neural cells in an undifferentiated state by regulating NR2E1.
Gas5	A3	Chr 5: Growth arrest-specific 5 has anti-tumor effects by inhibiting a number of miRNAs and lncRNAs, in part by functioning as a molecular sponge for some miRNAs. In cancer, it promotes apoptosis, inhibits proliferation, and invasion. GAS5 binds to the promoter of the insulin receptor gene and increases its expression. It directly interacts with EZH2 and also acts as a decoy for the glucocorticoid receptor (GR), blocking the upregulation of gene expression by activated GR.
1110038B12Rik	A3	Chr 17: RIKEN cDNA 1110038B12 aka Atrolnc-1 is abundantly expressed in skeletal muscles, and its expression is markedly increased in atrophying muscles by interacting with ABIN-1, an endogenous NF-κB inhibitor, leading to increased MuRF-1 expression and muscle proteolysis.

## 3. Discussion

Our results are consistent with several earlier studies that have identified ribosome-bound lncRNAs using the TRAP method, and as we report here, a dynamic regulation of this attachment due to external stimuli [19,27]. Our RNA-Seq results indicate that the attachment of nearly a third of the lncRNAs that we were able to detect are altered by EtOH. The addition of a PARP inhibitor to this EtOH treatment reduces the number of lncRNAs attached to ribosomes (by EtOH alone) by approximately half.

Alcohol’s effects on the very process of translation have been reported in both bacteria [32] and rodents [33]. In these studies, EtOH has an adverse mechanistic effect on ribosome processivity as well as protein translation. Consequently, translating ribosomes with RNA transcripts may comprise a uniquely selected subsample of the larger transcriptome that is additionally regulated by EtOH. Furthermore, PARP-1 has an important role in ribosomal biogenesis through several mechanisms. PARP-1 has a variety of effects on gene regulation, including adding ADP-ribose groups to histone proteins and participating in DNA demethylation. PARP-1 deletion mutants manifest delays in rRNA processing and an increase in rRNA intermediates [34]. Additionally, PARP can bind to RNA molecules, and conversely, RNA can activate PARP catalytic activity [35]. Finally, PARP enzymes are now shown to ADP-ribosylate the phosphorylated ends of both DNA and RNA molecules [36,37], and although demonstrable by *in vitro* experiments, it is unknown whether this function has consequences for ribosomal attachment *in vivo*.

Of the eight lncRNA molecules we validated, growth arrest-specific 5 (Gas5) is particularly well studied. GAS5 is associated with distinct biological functions, being regulated by the NMD pathway, and in turn targets the activity of apoptosis genes. Gas5 has been well characterized and exemplifies possible roles for lncRNAs beyond protein translation. GAS5 also participates in heterochromatin formation and can help recruit the polycomb repressive complex 2 (PRC2) to particular genomic locations to participate in heterochromatin formation [10]. Besides recruiting H3K4me4 histone methyltransferases and the H3K27me3 demethylase UTX (KDM6A), GAS5 can also act as a decoy nucleotide binding site for the glucocorticoid receptor, implicating it in the etiology of psychiatric illnesses via this glucocorticoid mechanism [38]. GAS5 biological function is partly dependent on highly conserved small nucleolar RNAs (snoRNAs) that are harbored within its introns. These actions modify chromatin by inducing active chromatin through increased H3K4me3 and reduced H3K27me3 [39]. We have independently measured GAS5 in PBMC samples of psychiatric patients and noted both a diagnostic effect as well as regulation by the use of antipsychotic medications [10].

Furthermore, 5430416N02Rik (also known as Gm3514 and Adapt33) acts as a scaffolding molecule to aid chromatin architectural proteins (CAPs), such as CTCF and KLF4, in reshaping chromatin architecture. In fact, 5430416N02Rik also has been shown to regulate the proliferation of embryonic stem cells by interacting with Mid1 loci (Mid1 promotes proliferation of ESCs), further demonstrating its function in mediating chromatin interaction [40].

Moreover, 1110038B12Rik (also known as Atrolnc-1) promotes proteolysis without affecting protein synthesis. At the molecular level, it interacts with ABIN-1, an endogenous NF-κB inhibitor [41]. Snhg is a recently identified family of lncRNA molecules (small nucleolar RNA host gene (SNHG)) whose family members are considered oncogenes in several cancers. We validated three members of this family, Snhg1, Snhg12, and Snhg17, which have been reported as stable biomarkers that are evolutionarily conserved [42]. Snhg1 has previously been identified as a ribosome-bound lncRNA [30] and may play an important role in microglia activation by sequestering miR-329-3p and modulating its activity as well as affecting Wnt/β-catenin and PI3K/AKT/mTOR signaling pathways. Snhg12 is expressed in the nucleus, is dynamically regulated, does not encode short peptides in peripheral macrophages, and is polyadenylated [42]. Snhg12 interacts with DNA-dependent protein kinase (DNA-PK), an important sensor/mediator in the DNA damage response, and facilitates DNA-PK activity and, conversely, a reduction in Snhg12 will increase DNA damage [42]. Snhg12 can function in an RNA network comprised of lncRNA-miRNA-mRNA interactions, with the lncRNA serving as a miRNA sponge that regulates the target protein-coding mRNA [43]; this ‘competing endogenous RNA’ or ‘ceRNA’ is protective in alcohol-induced esophageal carcinoma. Du et al. identify several such ceRNA networks involving Snhg12, specifically SNHG12–miR-1–ST6GAL1 and SNHG12–hsa-miR-33a–ST6GAL1, elucidating the regulatory role of hsa-miR-1 and miR-33a in modulating the protein ST6GAL1 (β-galactoside α2,6 sialyltransferase 1) [43].

Lastly, we also validated two host genes for miR124 and miR9: lncRNAs MIR124-HG and MIR9–3HG. miR-124 and miR-9 both play an important role in neuronal development and are abundantly expressed brain-specific miRNAs [44,45]. miR-124 regulates neuronal differentiation during brain development and plays an important role in neurogenesis and neuronal function [46,47]. Additionally, miR-124 was shown to play a role in the pathophysiology of neuropsychiatric disorders, such as Alzheimer’s disease and autism [47,48]. MicroRNAs are associated with ribosomal structures, particularly the 28S subunit, and this association occurs in both the nucleolus as well as the cytoplasm [49]. miRNA localization at both these locations has been implicated in the early targeting of the nascent mRNA transcript and continued association into the cytoplasmic ribosome where translation is regulated.

In summary, the current study indicates that EtOH by itself can significantly affect lncRNA association with ribosomes, perhaps affecting their ability to code for small peptides or their role in regulating translation. In addition, this effect is significantly modified by the co-administration of a PARP inhibitor, an effect that could have therapeutic implications.

## 4. Materials and Methods

### 4.1. Transgenic Mice

To obtain total RNA for TRAP, we used a mouse strain in a C57BL/6J background that expresses a fluorescently tagged ribosomal protein (EGFP-Rpl10a) under a CaMKIIα promoter. These mice are derived by crossing C57BL/6J-Tg(tetO-EGFP/Rpl10a)5aReij/J with B6. Cg-Tg (CaMKIIα-tTA)1Mmay/DboJ (CaMKIIα-tTA) mice were used. In the cortex, CaMKIIα is expressed primarily in cortical pyramidal neurons [50,51]. Both mouse lines were purchased from Jackson Laboratories. The resulting offspring have EGFP-tagged ribosomes expressed only in CaMKIIα-positive cells [27,28]. The genotyping of mice was performed according to standard procedures.

### 4.2. Animal Treatment and Sample Collection

All animal studies were conducted in accordance with the National Institutes of Health Guidelines for the Care and Use of Laboratory Animals. Male transgenic mice that were 8–12 weeks old were randomly assigned to four treatment groups: saline (CTL), ethanol (EtOH), ABT-888 (ABT), and ethanol + ABT-888 (EtOH+ABT). The animals were administered i.p. twice a day with normal saline (CTL) or EtOH (final daily dose of 2 g/kg) for four consecutive days (two hours between the first and second injection of the day). ABT-888 (25 mg/kg) was co-administered on the fourth day in a sub-group of mice that received saline (ABT treatment only) and EtOH (EtOH+ABT) the previous three days (a schematic diagram is shown in Figure 5). The dosing strategy was the same as the involuntary ethanol administration protocol reported earlier [6]. The time point of ABT-888 administration was chosen because the half-life of ABT-888 is 1.2–2.7 h with a Tmax of 20 min, thereby providing the longest possible duration of ABT-888 exposure during the experiment. Also, in our earlier study, we observed a change in drinking behavior after a single dose of ABT-888 given after a period of 4 days of EtOH administration. We simulated the drinking-in-the-dark binge drinking paradigm (this is a 4 day drinking paradigm) in which the PARP inhibitor reduced drinking. Two hours after the second injection on day 4, mice were sacrificed via CO_2_ and decapitation. Blood samples were analyzed for BEC, and the brain was dissected on a brain block to isolate the PFC based on the coordinates from our earlier publication [6]. Tissue was homogenized and subjected to immunoprecipitation-driven RNA extraction according to the TRAP protocol. The experiment was designed to include extra animals in each group for future validation; however, the number of samples utilized for RNA-seq was based on logistics and expense. Samples sent for RNA-Seq included the following: CTL (*n* = 3), EtOH (*n* = 4), and EtOH+ABT-888 (*n* = 4).

**Figure 5 ncrna-09-00072-f005:**
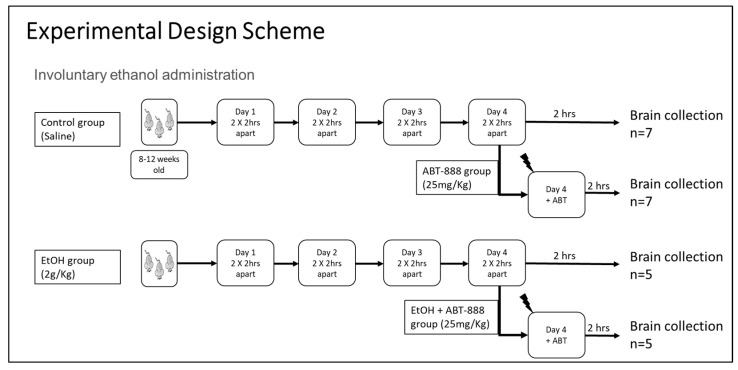
Schematic diagram of the experimental design demonstrating the different groups used and the number of samples collected. Adult C57BL/6J (8–12 weeks) male transgenic mice were group housed on a standard 12 light/12 dark cycle and allowed ad libitum access to food and water. Mice were subjected to a fixed-dose regimen of EtOH [52].

### 4.3. Blood Ethanol Concentration (BEC)

An enzymatic assay calibrated against a standard curve of EtOH known concentrations was utilized to determine BECs [53], as previously described [7]. BEC was calculated in milligrams (mg) per deciliter (dL) and noted as follows: CTL group (13.2 ± 18.9 mg/dL; *n* = 7), EtOH group (285 ± 84.2 mg/dL; *n* = 5), and EtOH+ABT group (221 ± 113 mg/dL; *n* = 5).

### 4.4. Immunoprecipitation and RNA Extraction (TRAP Protocol)

The technology utilized was developed in the Heintz lab at Rockefeller University [27]. Briefly, PFC samples were homogenized in the appropriate buffer (containing NP-40, KCl, Tris, MgCl2, cycloheximide, protease inhibitor cocktail, RNase inhibitors, and DTT) and centrifuged. 40 µL of supernatant were collected for total RNA (labeled as ‘input’), and the remaining supernatant (approximately 400 µL) was transferred to a separate tube and treated with anti-EGFP antibodies and Agarose Plus A/G beads (Santa Cruz_sc-2003) overnight. After centrifugation, the supernatant with beads containing ribosome-bound RNA (TRAP samples) was processed for RNA elution and extraction using QIAZOL (Qiagen # 79306). TRAP-RNA and total-RNA samples were finally extracted using an miRNeasy Mini Kit (Qiagen # 217004).

### 4.5. RNA-Seq Sample Preparation and Analysis

RNA samples were quantified using a Quantus fluorimeter (Promega, Madison, WI, USA) using dual RNA/DNA quantification. Levels of remaining DNA did not exceed 10% of the total amount of Nucleic Acid. RNA integrity numbers (RINs) (Mean:7.33; SD: 0.46; Min: 6.4; Max: 7.8) were assessed using an Agilent 4200 TapeStation (Agilent, Santa Clara, CA, USA) (Mean:7.33; SD: 0.46; Min: 6.4; Max: 7.8). The samples were subjected to rRNA depletion, and sequencing libraries were prepared for Illumina sequencing using a CORALL Total RNA-Seq Library Prep Kit with Unique Dual Indices (Lexogen, Vienna, Austria, PN M11696) with an RiboCop HMR rRNA Depletion Kit V1.3 (Lexogen, PN K03796). Final amplified libraries were purified and quantified, and the average fragment sizes were confirmed to be 254bp to 323bp and then subjected to test-sequencing on an MiniSeq instrument in order to check sequencing efficiencies. Sequencing was carried out on a NovaSeq 6000 (Illumina, San Diego, CA, USA) with SP flowcell and 2 × 50 bp PE reads.

The resulting FASTQ files were checked for adapters, and following QC, raw reads were aligned to the reference genome, mm10, in a splice-aware manner using the STAR aligner [54]. The genome reference utilized the ENSEMBL gene and transcript annotations, which included non-coding RNAs and mRNAs. The expression level of the genes was quantified using FeatureCounts [55]. Differential expression statistics (Log2 fold change and *p*-value) were computed using edgeR [56,57] on raw expression counts obtained from quantification. Prior to analysis, the counts were filtered to exclude any gene that either had less than a total of 500 counts across all samples or was detected in fewer than 3 samples; we eliminated any lncRNAs that were poorly detected (given the low expression of lncRNAs) to increase the rigor of analysis. The data were normalized using the trimmed mean of m-values (TMM) normalization. The normalized data were modeled across treatment status, i.e., CTL, EtOH, or EtOH+ABT-888. The terms of the model were tested using the likelihood ratio test, i.e., glmLRT function, within edgeR. Pairwise tests of the expression data were computed using the exactTest function in edgeR. A principal component (PCA) analysis was performed using the R analysis of PCA and Scree plots for clustering the samples based on gene expression patterns in order to examine the level of similarity or dissimilarity in the gene expression profiles of the three experimental groups. We performed comparisons entirely within the TRAP samples as follows: CTL_TRAP (*n* = 3); EtOH_TRAP (*n* = 4); and EtOH+ABT_TRAP (*n* = 4). The entire dataset utilized in both studies, including the processed files, is accessible in the GEO repository under accession number GSE227947.

### 4.6. In Silico Comparison of Protein-Coding Potential Identities of Ribosome-Attached lncRNAs

We examined protein-coding potential in our subset of ribosomal lncRNAs using the *in silico* method of Zeng et. al., 2018. Analyzing multiple human and mouse datasets from various tissues, the authors categorize ribosomal lncRNAs based on their ability to translate proteins/peptides. LncRNAs with a putative ORF are processed through three coding filters using ribosomal footprints (FLOSS), ribosome release (RRS), and the reading frame from three-base periodicity (triplet periodicity, Framescore). A summary of the results of these three filters comprises the ‘Ribosomal Affiliation Index’ (RAI) and has the following three values: “A0”, passed NO translation filter; “A1”, passed ONE filter; “A2”, passed TWO filters; and “A3”, passed THREE filters. LncRNAs labeled “N” were not expressed in the dataset presented by Zeng et. al. We cross-referenced this additional annotation to our list of 308 lncRNA identities and obtained 30 lncRNA molecules that could be tagged with an A0 to A3 and N label (Figure 6).

### 4.7. Quantitative Real-Time Reverse Transcription Polymerase Chain Reaction-Seq Sample Preparation and Analysis

Target gene expression was normalized to the geometric mean of Hprt and Actb. For qRT-PCR, cDNA samples were analyzed using PikoReal Real-Time PCR (Thermo Fisher, Waltham, MA, USA) and Maxima^®^ SYBR Green/ROX qPCR Master Mix (Thermo Fisher, Waltham, MA, USA). Primers were designed to span at least one intron–exon boundary, and amplicons were tested using melting curve analysis. The following cycling conditions were used: 10 min at 95 °C, then 40 cycles at 95 °C for 15 s, 60 °C for 20 s, and 72 °C for 20 s. Gene validation by qRT-PCR was analyzed with a one-way analysis of variance (ANOVA) followed by post hoc Bonferroni or Tukey tests or with Student’s *t*-test, when appropriate. All statistical analyses were performed using GraphPad Prism version 7 for Windows (GraphPad Software, San Diego, CA, USA) or SPSS (IBM Corp. released 2016, IBM SPSS Statistics for Windows, version 24.0, Armonk, NY, USA).

### 4.8. Validation of Translating RNA from CaMKIIα-Expressing Cell Isolation Using Gene-Specific Expression

We assessed various markers to confirm the successful isolation of RNA from CaMKIIα-expressing cells. Specifically, we quantified Egfp, CamKIIα, S100b, and Ef101557 expression levels in TRAP samples. Isolated ribosomes will have higher expression of green-fluorescent protein (Egfp) and CamKIIα as well as a low expression of the astrocyte-specific S100b and microglial-specific Ef101557 transcripts. Our results strongly support the efficacy of the TRAP technique in isolating CaMKIIα-expressing cells, as indicated by the elevated levels of Egfp (t6 = 2.547, *p* = 0.04, *n* = 4 per group) and CamKIIα (t4 = 6.022, *p* = 0.004, *n* = 3 per group) compared to total RNA, while S100b (t6 = 3.357, *p* = 0.02, *n* = 4 per group) and Ef101557 (t6 = 6.063, *p* = 0.0009, *n* = 4 per group) were reduced compared to total RNA.

## 5. Limitations

The number of tissue samples was a limiting factor given this is a genome-wide study, and this perhaps limited our discovery of additional ribosome-attached lncRNA molecules impacted by EtOH. Low-expression lncRNAs (naturally lowly expressed) may not be present, and in addition, the capture by ribosomes is not as vigorous as with mRNA. Consequently, in our genome-wide sequencing effort, we could consider only 308 lncRNAs to present in our results. Finally, given the very early stage in lncRNA research, the functional, structural, and network characteristics of these molecules are in a very early stage. Consequently, our ability to select molecules for validation is also limited by the *in silico* analysis presented by Zeng et al., 2018 [18], as noted here.

## Figures and Tables

**Figure 1 ncrna-09-00072-f001:**
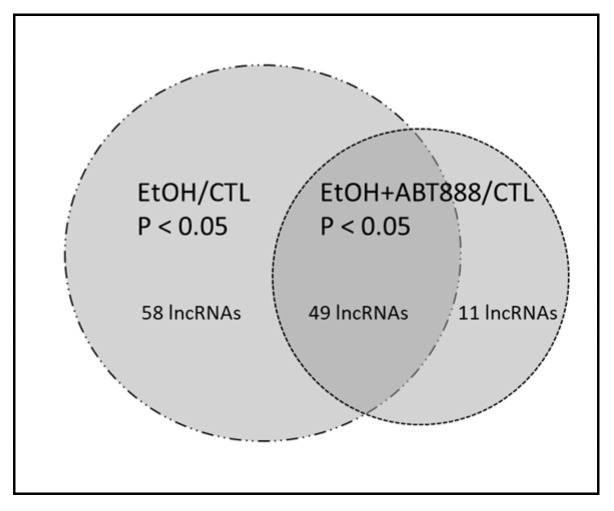
Venn diagram illustrating the differentially expressed transcripts identified in the two comparisons identified by TRAP-Seq. EtOH resulted in the differential expression of 107 (58 + 49) lncRNAs (EtOH vs. CTL; FDR < 0.05) represented by the “dash line” circle. Co-administering the PARP inhibitor ABT-888 (EtOH+ABT-888) resulted in 60 (49 + 11) differentially altered lncRNAs (EtOH+ABT-888 vs. CTL; FDR < 0.05) represented by the “dotted” circle. Out of these 60 lncRNAs, 11 were uniquely altered by EtOH+ABT-888 but not EtOH, whereas 49 lncRNAs were altered by both EtOH and EtOH+ABT-888.

**Figure 2 ncrna-09-00072-f002:**
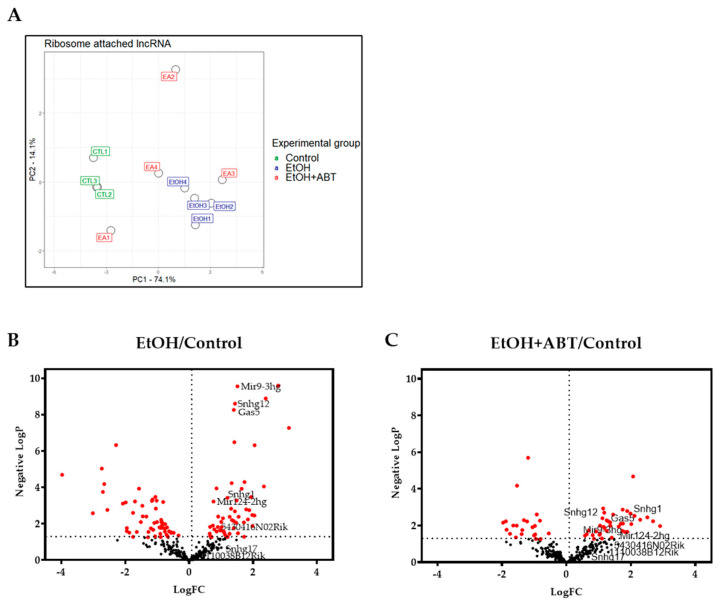
Systematic evaluation of the TRAP-seq data. (**A**) PCA plots of PC1 vs. PC2 on the log CPM of TRAP-seq data, illustrating the characteristics of samples according to expression profiles. As a result, the first two PCs explained more than 74% of the variability among the sample. Each dot indicates a sample, CNT (*n* = 3), EtOH (*n* = 4), and EtOH+ABT (*n* = 4). (**B**) Volcano plots displaying the distribution of lncRNAs attached to ribosomes after the administration of EtOH and EtOH+ABT compared to control. Changes in attachment to ribosomes are seen through the distribution of the dots (individual lncRNAs). The eight lncRNAs chosen for validation are labeled and their changes can be followed in both plots, (**B**) effect of EtOH and (**C**) effect of EtOH+ABT, on the attachment of lncRNAs to ribosomes compared to the CTL. The volcano plot indicates -log10 P-value (*Y*-axis) plotted against their respective log2-fold change (*X*-axis). Positive FC represents increased attachment of lncRNAs, while negative FC represents a reduced attachment of lncRNAs between groups. The horizontal line represents −log (0.05) = 1.3, and the vertical lines represents FC = 0.

**Figure 3 ncrna-09-00072-f003:**
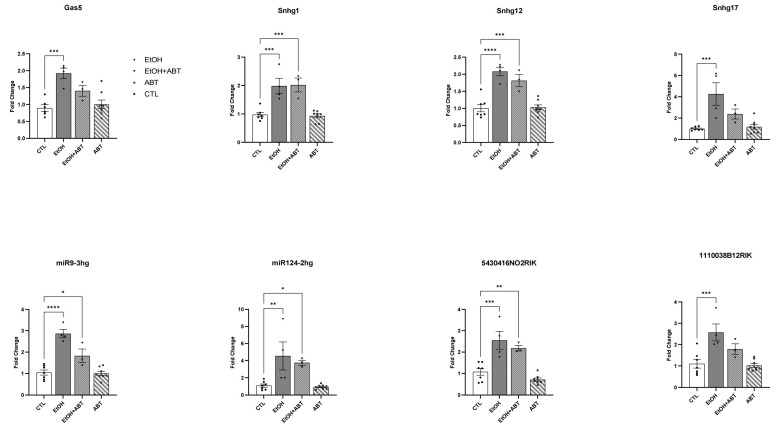
The relative expression levels of eight selected lncRNAs validated by qRT-PCR in four experimental groups: CTL, EtOH, EtOH+ABT, and ABT. Data are expressed as fold change (FC) ± S.E.M. (error bars) after normalization to the geometric means of internal controls. CTL, filled circle; EtOH, rhombus; EtOH+ABT, inverted triangle; ABT, circle. *p*-value * < 0.05, ** < 0.01, *** < 0.001, **** < 0.0001 compared to the CTL group, as analyzed by one-way ANOVA.

**Figure 4 ncrna-09-00072-f004:**
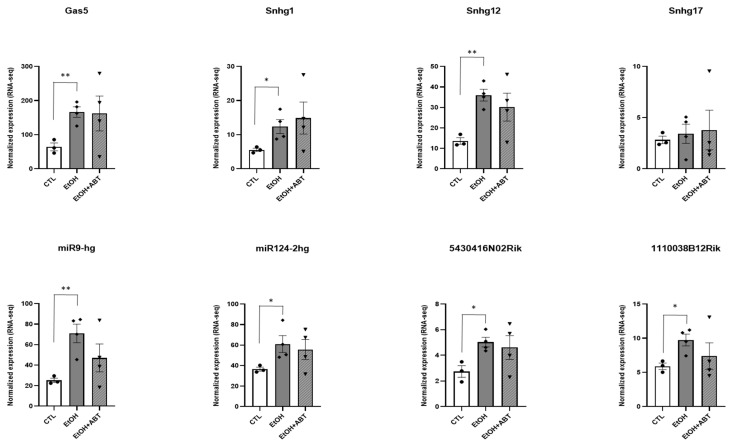
TRAP-Seq data showing expression levels of the eight selected lncRNAs. Normalized expression levels of the lncRNAs in CTL, EtOH, and EtOH+ABT derived from sequencing results. CTL (*n* = 3), EtOH (*n* = 4), EtOH+ABT-888 (*n* = 4). CTL, filled circle; EtOH, rhombus; EtOH+ABT, inverted triangle *p*-Value * < 0.05, ** < 0.01 compared to the control group, as analyzed by one-way ANOVA.

**Figure 6 ncrna-09-00072-f006:**
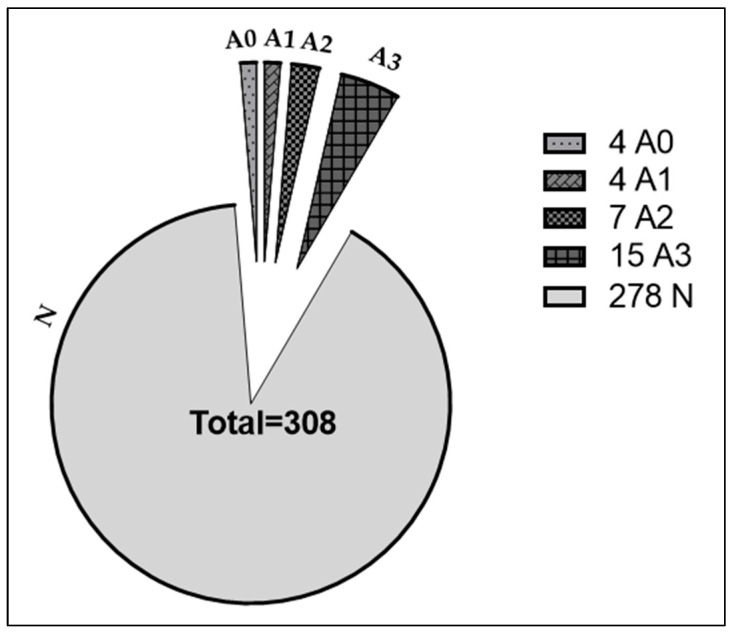
Classification of identified lncRNAs from TRAP-Seq data based on the *in silico* method of determining the protein-coding potential described in Zeng et al. 2018. The distributions of differentially expressed lncRNAs are shown in the legend as A0, A1, A2, A3, and N. Of the eight lncRNAs validated, Mir124-2hg and 5430416NO2Rik are classified as A0. Snhg17 is A2, and Snhg12, Snhg1, Mir9-3hg, Gas5, and 1110038B12Rik are A3.

## Data Availability

The data presented in this study are openly available in the GEO repository, accession number GSE227947.

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
