# Peer review of "Ethanol- and PARP-Mediated Regulation of Ribosome-Associated Long Non-Coding RNA (lncRNA) in Pyramidal Neurons"

_ncrna, 2023, doi:10.3390/ncrna9060072_

Round 1

Reviewer 1 Report

Comments and Suggestions for Authors

Rizavi and colleagues present a study describing the effects of alcohol and PARP inhibitor on the lncRNA ribosomal binding. The study has merit; however, the research design is weak, and the underlying hypothesis justifying this study is lacking. My comments are presented below. 

1. My first comment is conceptual. The authors mention the use of the involuntary DID paradigm to, as it seems, test the ABT inhibitor effect on binge drinking. However, this is not well explained. The choice of introducing alcohol via intraperitoneal injections is also not very well explained, which, while on its own, is not wrong, the authors should not have justified why they used the intraperitoneal route of introducing alcohol rather than a different paradigm, such as intermittent alcohol protocols to measure voluntary alcohol consumption. Such paradigms are also better suited to provide a neurobehavioral link between the observed molecular mechanism and alcohol drinking measures. 

2.  My second conceptual concern is the exclusive focus on pyramidal neurons. Again, while this is not necessarily wrong, it would have been helpful if the authors had also done a TRAP assay in cell types other than neurons or even in different brain regions, e.g., the amygdala. 

3. It was not well explained why the control (CTL) group had such a high BEC concentration. 

4. Some GWAS results, e.g., Gelernter et al. (2014), have also implicated a lncRNA locus on chromosome 4 potentially associated with alcohol dependence. Were any of the lncRNA identified by the Gelernter et al. study also present in the authors' list of lncRNA?  

Author Response

  1. My first comment is conceptual. The authors mention the use of the involuntary DID paradigm to, as it seems, test the ABT inhibitor effect on binge drinking. However, this is not well explained. The choice of introducing alcohol via intraperitoneal injections is also not very well explained, which, while on its own, is not wrong, the authors should not have justified why they used the intraperitoneal route of introducing alcohol rather than a different paradigm, such as intermittent alcohol protocols to measure voluntary alcohol consumption. Such paradigms are also better suited to provide a neurobehavioral link between the observed molecular mechanism and alcohol drinking measures.

We have previously worked with the drinking-in-the-dark (DID) paradigm (Vallerini et al 2020). In this study our methods were designed to examine the effects of a specific dose of ethanol on ribosomal attachment. This is the same approach utilized in our previously published report (Krishnan et al., 2023) that generated the TRAP-seq data which was analyzed for differential expression of coding genes. Here we analyzed, from the same data set, differential expression of lncRNAs attached to ribosomes. We considered our current dosing strategy for two reasons. We were trying to simulate the DID paradigm (this is a 4-day drinking paradigm) in which PARP inhibitor reduced drinking. Primarily, the time point of ABT-888 administration was chosen because the half-life of ABT-888 is 1.2–2.7 h, with a Tmax of 20 min, thereby providing the longest possible duration of ABT-888 exposure during the experiment.  Also, in our earlier study (Vallerini et al., 2020), we observed a change in drinking behavior after a single dose of ABT-888 given after a period of 4 days of EtOH drinking behavior. These initial experiments were designed to avoid behavioral or contingent variability and sought to establish a primary pharmacological response of ethanol on ribosome attachment. We have clarified the connection in the introduction and further explained our rationale in the “Methods” section 4.2.

2.  My second conceptual concern is the exclusive focus on pyramidal neurons. Again, while this is not necessarily wrong, it would have been helpful if the authors had also done a TRAP assay in cell types other than neurons or even in different brain regions, e.g., the amygdala.

We appreciate the feedback concerning the focus on pyramidal neurons in our study and recognize the potential benefits of exploring the cellular and regional specificity of our findings. However, it's important to note that due to the novel TRAP paradigm employed in our study and the associated costs, our primary focus was placed on pyramidal neurons. Our reasoning stems from our earlier work, where we demonstrated that virally induced PARP1 overexpression in the PFC increased voluntary alcohol consumption while the PARP inhibitor ABT-888 decreased it (Vallerini et al., 2020). The cell-type specificity of the study also derives from the availability of a transgenic mouse model that expresses a GFP-Rpl10a ribosomal protein under the control of the promoter for CaMKIIα, an enzyme expressed predominantly in excitatory pyramidal neurons.

3. It was not well explained why the control (CTL) group had such a high BEC concentration.

We appreciate your concern. The CTL group served as a baseline for comparative analysis alongside the experimental group. It's important to note that the BEC assay itself introduces inherent individual variation, as evidenced by the range of BEC values recorded in the control group (13.2 ± 18.9 mg/dl), as well as in the experimental groups, the EtOH group (285 ± 84.2 mg/dl; n = 5) and the EtOH + ABT-888 group (221 ± 113 mg/dl; n = 5). BEC was calculated to ensure that the experimental group achieved BECs greater than 100 mg/dl and to exhibit evidence of intoxication Finally, to safeguard against this experimental variability, we have normalized both experimental conditions to the control condition (EtOH/Control and EtOH+ABT/Control).

4. Some GWAS results, e.g., Gelernter et al. (2014), have also implicated a lncRNA locus on chromosome 4 potentially associated with alcohol dependence. Were any of the lncRNA identified by the Gelernter et al. study also present in the authors' list of lncRNA?

Thank you for your insightful question regarding the potential overlap between our identified long non-coding RNAs (lncRNAs) and those highlighted in the GWAS study by Gelernter et al. (2014). Gelernter et al. (2014) observed associations of SNP rs116203444 across the LOC100507053 locus, which codes for a lncRNA, in African-American population with alcohol dependence.  We have examined our data set of lncRNAs and can confirm that this lncRNA was not found among the lncRNAs listed in our research.

Reviewer 2 Report

Comments and Suggestions for Authors

The article "Ethanol and PARP-mediated Regulation of Ribosome-Associated Long Non-Coding RNA (lncRNA) in Pyramidal Neurons" presents an intriguing investigation into the potential translation of lncRNAs associated with ribosomes in the context of ethanol exposure and PARP inhibition. It comprehensively analyzes the effects of ethanol and ABT-888 on ribosomal attachment of lncRNAs, with clear differentiation between the two conditions. Additionally,

Minor comments:

1.     It might be beneficial to briefly mention the significance of ribosome-bound lncRNAs and their potential implications in the introduction.

2.     A brief explanation of the rationale behind selecting the eight lncRNAs for qRT-PCR validation would provide context for the validation process.

3.     In the discussion section, it would be beneficial to discuss the broader implications of the study's findings in the context of neurobiology and potential therapeutic applications. Highlighting the clinical relevance of these findings could engage readers and underscore the importance of further research in this area.

Overall, the article offers valuable insights into the intricate relationship between ethanol exposure, ribosomes, and lncRNAs. Addressing the suggested improvements could strengthen the presentation and impact of the study's findings.

Comments on the Quality of English Language

English is fine, can be improved.

Author Response

1. It might be beneficial to briefly mention the significance of ribosome-bound lncRNAs and their potential implications in the introduction. It might be beneficial to briefly mention the significance of ribosome-bound lncRNAs and their potential implications in the introduction.

Thank you for your comment. In the introduction, we have clarified our current understanding of the implications of ribosomal attachment to lncRNAs. Experimental and in silico analyses have reported that approximately 50% of lncRNA have a ribosome footprint. Furthermore, increasing evidence of small peptide translation, particularly of evolutionary conserved transcripts, as well as trans-regulation of coding transcripts, implies another layer of gene regulation.

2. A brief explanation of the rationale behind selecting the eight lncRNAs for qRT-PCR validation would provide context for the validation process.

Thank you for your feedback and suggestions. To provide context for our validation process, we'd like to highlight the "Methods" section (4.6) and the "Results" section (2.2). We employed an approach based on the analysis conducted by Zeng et al. (2018) across multiple human and mouse datasets. Utilizing the analysis by Zeng et al (2018) we matched mouse genes that overlapped our set of ribosome-attached lncRNAs, providing 30 lncRNAs common to both datasets. Of these, we identified eight lncRNAs that displayed statistically significant alterations (t-test) in expression levels due to ethanol administration. These eight lncRNAs were subsequently chosen for qRT-PCR validation, thereby validating the RNA-seq results.

3. In the discussion section, it would be beneficial to discuss the broader implications of the study's findings in the context of neurobiology and potential therapeutic applications. Highlighting the clinical relevance of these findings could engage readers and underscore the importance of further research in this area.

Thank you for your valuable feedback. Here we report results of lncRNA’s that have not been fully annotated and their attachment to ribosomes, whose function is primarily ‘translating’ gene-coding RNA transcripts. We have thus hesitated to speculate on potential clinical or therapeutic implications.

Reviewer 3 Report

Comments and Suggestions for Authors

Hooriyah et al. in their manuscript entitled, "Ethanol and PARP-mediated regulation of ribosome associated long non-coding RNA (lncRNA) in pyramidal neurons" have explored the intricate relationship between alcohol consumption and gene regulation, with a particular focus on the impact of alcohol use disorder (AUD) and the potential role of long non-coding RNAs (lncRNAs) in the brain. AUD affects a substantial portion of adults in the United States, prompting extensive investigation into its genetic underpinnings. Previous studies have highlighted the significance of prefrontal cortex neurons in the reinforcing effects of alcohol. Their previous study has also demonstrated that inhibiting Poly-ADP Ribose Polymerase (PARP) enzymatic activity can reverse gene expression changes caused by alcohol administration and reduce alcohol-drinking behavior in mice. Moreover, recent investigations have explored the effects of ethanol (EtOH) and PARP on the attachment of ribosomes to mRNA transcripts of protein-coding genes. By utilizing ribosome purification in conjunction with unbiased RNA sequencing (known as TRAP-Seq, or Translating Ribosome Affinity Purification followed by RNA sequencing), researchers can pinpoint transcripts that are intricately associated with ribosomes. In this study, the focus expands to encompass the effects of EtOH and PARP inhibition via ABT-888 on lncRNAs in the brain. The study utilizes transgenic mouse lines to investigate the effects of EtOH and PARP inhibition on lncRNA attachment to ribosomes. Results indicate that EtOH alone significantly affects lncRNA attachment to ribosomes, with the addition of a PARP inhibitor mitigating this effect. The research suggests that alcohol's impact on translation is complex and may involve a select subset of the transcriptome. The study delves into specific lncRNA molecules, highlighting their roles and interactions. For instance, Gas5 is associated with heterochromatin formation and apoptosis gene activity regulation. The lncRNA 5430416N02Rik plays a scaffolding role in chromatin architecture, while 1110038B12Rik is linked to proteolysis. The Snhg family of lncRNAs has implications in cancer, and their interactions with miRNAs and mRNAs are explored. Additionally, validated host genes for miR124 and miR9 are discussed, given their relevance to neuronal development and function.

In conclusion, this study sheds light on how alcohol affects the association of lncRNAs with ribosomes, potentially influencing their regulatory functions and peptide-coding abilities. The co-administration of a PARP inhibitor modifies these effects, suggesting therapeutic possibilities. This research contributes to a deeper understanding of the intricate mechanisms underlying gene regulation in the context of alcohol consumption and its potential implications for neuropsychiatric disorders. I think this work is of significance and would be of great value to the readers of ncRNA.

Author Response

In conclusion, this study sheds light on how alcohol affects the association of lncRNAs with ribosomes, potentially influencing their regulatory functions and peptide-coding abilities. The co-administration of a PARP inhibitor modifies these effects, suggesting therapeutic possibilities. This research contributes to a deeper understanding of the intricate mechanisms underlying gene regulation in the context of alcohol consumption and its potential implications for neuropsychiatric disorders. I think this work is of significance and would be of great value to the readers of ncRNA.

Thank you for your insightful feedback and acknowledgment of the significance of our research. We're delighted that you find our study significant and of potential value to ncRNA readers. This research indeed aims to uncover the intricate mechanisms at the intersection of alcohol consumption, lncRNAs, and ribosomal associations, potentially offering insights into therapeutic applications. We sincerely appreciate your thoughtful evaluation of our study, and your positive feedback motivates us to continue this important work.

Round 2

Reviewer 1 Report

Comments and Suggestions for Authors

I have no more comments for the authors.